# Coaching via Telehealth: Caregiver-Mediated Interventions for Young Children on the Waitlist for an Autism Diagnosis Using Single-Case Design

**DOI:** 10.3390/jcm10081654

**Published:** 2021-04-13

**Authors:** Megan G. Kunze, Wendy Machalicek, Qi Wei, Stephanie St. Joseph

**Affiliations:** Department of Special Education and Clinical Sciences, University of Oregon, Eugene, OR 97403, USA; wmachali@uoregon.edu (W.M.); qwei4@uoregon.edu (Q.W.); sstjose2@uoregon.edu (S.S.J.)

**Keywords:** autism spectrum disorder, restrictive and repetitive behavior, caregiver-mediated intervention, applied behavior analysis, telehealth, single-case design

## Abstract

Years can elapse between parental suspicion of a developmental delay and a diagnostic assessment, ultimately delaying access to medically necessary, autism-specific intervention. Using a single-case, concurrent multiple baseline design, autism spectrum disorder symptomology (i.e., higher-order restrictive and repetitive behaviors and interests; higher-order RRBIs) was targeted in toddlers (21–35 months) waiting for a diagnostic appointment. Caregivers were coached via telehealth to mediate early intervention to decrease interfering, inflexible higher-order RRBIs during play using four evidence-based applied behavior analytic strategies: modeling, prompting, differential reinforcement of appropriate behaviors, and response interruption and redirection. Six mother–child dyads were recruited from pediatrician offices and early intervention service districts in the United States. All families were considered under-served, under-resourced, or living in rural locations. A visual analysis of the data combined with Tau-U revealed a strong basic effect between the intervention package and parent strategy use and child flexible and inflexible behavior. Findings were consistent across participants with one exception demonstrating a moderate effect for flexible behaviors yet a strong effect for inflexible behaviors. Standardized mean difference was beyond zero for all participants. Implications for science and practice include support for early intervention of higher-order RRBIs for young children with and at risk for ASD.

## 1. Introduction

### 1.1. Waitlists for Autism Diagnosis

Autism spectrum disorder (ASD), an early emerging neurodevelopmental disorder defined by social communication delays and restricted and repetitive behavior and interests (RRBIs) [1], has steadily risen to the current rate of 1 in 54 children in the United States (US) [2] and 1 in 160 children worldwide [3]. The increase in children at risk for autism has caused a surge in wait times for a medical diagnosis. Although statistics vary by country, long waitlists specific to an ASD diagnosis are not unique to the US (e.g., in Ireland [4]; in the UK [5]; in Australia [6]; in Canada [7]). Years can elapse between parental suspicion of a developmental delay and a diagnostic assessment, ultimately delaying access to medically necessary, autism-specific intervention [8]. Early diagnosis of ASD is imperative as individualized early intervention (e.g., applied behavior analysis; ABA) can reduce the severity of ASD symptomology and, for some, result in no longer meeting diagnostic criteria for ASD [9]. Referrals often occur before age two, and statistics demonstrate that children can receive a reliable diagnosis by 18 months (e.g., Autism Diagnostic Observational Schedule (ADOS-2) Toddler Module) [10,11]. Yet, the average age for diagnosis in the United States is age 4 [8]. For children of color who live in rural US locations, the average age is even later: five years old and beyond [8,12,13].

In addition, long waitlists and inaccessible services negatively impact parents’ well-being by increasing their stress and anxiety while decreasing their sense of efficacy as a parent due to limited information regarding diagnostic processes [4], inadequate skills to respond to challenging behaviors [5], and limited intervention resources [7]. Barriers to service utilization include difficulty finding medical providers, extended travel to reach qualified providers, and shortages of mental health professionals [13]. While early diagnosis of ASD prompts parents to pursue interventions beyond the more general early interventions provided pre-diagnosis [14], barriers persist. Streamlining procedures to ensure timeliness toward diagnosis is an obvious next step for the field [15,16,17,18]. Yet, early, ASD-specific intervention is imperative even while on a waitlist for diagnostic evaluation, and researchers should concurrently develop and evaluate the efficacy and feasibility of such interventions.

### 1.2. Early Intervention for ASD

Interventions for children with ASD commonly target core areas of support needs: social communication and RRBIs. RRBIs fall into two categories: lower and higher-order. Lower-order RRBIs are repetitive motor movements, including the stereotypical movement of objects in a repetitive manner (i.e., object not used in the manner intended by design) and automatic self-injurious movements conducted in a repeated sequence [1,15,16]. Higher-order RRBIs are cognitive behaviors such as inflexibility in routines and actions, the adherence to ritualistic series of events, and often include rigid, rule-bound actions [19,20]. There is controversy in addressing RRBIs overall [21] (e.g., intervening on non-socially significant behaviors is unnecessary). Yet, research suggests higher-order RRBIs are associated with heightened parent stress and more problem behavior than lower-order RRBIs [20]. Therefore differentiating subsets of RRBIs by the influence these behaviors have on the child’s and stakeholder’s quality of life is a crucial consideration in early intervention and research.

Early intervention packages (i.e., one or more intervention techniques used simultaneously or in tandem) that focus on social communication are common (e.g., Early Start Denver Model (ESDM); Milieu Training; Joint Attention Symbolic Play, Engagement, and Regulation (JASPER); More Than Words—Hanen Model) [22,23,24,25]. Although some packages have demonstrated mixed results (i.e., Hanen Model [25] and ESDM [22]), others are considered evidence-based practices with more promising outcomes (i.e., JASPER; Milieu Training) [26]. These early intervention packages focus on social communication rather than RRBIs despite the prevalence of both symptomologies, demonstrating interventions addressing RRBIs as an area of need for children with ASD. Additionally, higher-order RRBIs have a strong association with increased parental anxiety and low-self-efficacy; yet, higher-order RRBIs continue to be understudied in response to intervention [15].

Higher-order RRBIs can be particularly difficult for families of young children due to a child’s insistence on precise routines or interactions. Requests for a child to be more flexible (e.g., variation in play materials, partners, or scenarios) can trigger challenging behaviors [27]. These behaviors may escalate, potentially beginning with tantrums and leading to self-injury and harming others (e.g., head banging, biting, or property destruction) [27]. The presence of high levels of inflexible behaviors can prevent children from exploring play activities beyond their fixated interests, often rejecting a parent’s bids for interaction, resulting in isolation rather than social play and minimal reciprocation with family members [20]. While rigidity in routines and limited social responsivity are core characteristics of ASD, these behaviors may be present in toddlers and young children with various developmental delays. The commonality and difficulty caused by higher-order RRBIs make them plausible targets for early intervention across diagnostic labels. Importantly noted, preliminary research demonstrated that higher-order RRBIs could decrease with intervention [20].

Some interventions focusing on RRBIs do exist despite the comparatively fewer interventions than those addressing social communication. For example, interventions specific to lower-order RRBIs include such strategies as exercise [28] and environmental arrangement [29]. There is extensive research on interventions addressing self-injurious behaviors [30], which includes functional communication training [31], differential reinforcement [32], and response cost procedures [33]. Interventions for higher-order RRBIs, often using strategies from ABA, include a variety of antecedent and consequence interventions [34], such as differential reinforcement strategies, visual schedules [35], and video modeling [36].

In a study by Harrop and colleagues [37], the authors targeted parent responses to their toddler’s RRBIs during play scenarios. In their study of 83 toddlers, almost all displayed RRBIs during the parent–child play period. The most common RRBI was repetitive object use, thus highlighting the frequent occurrence of higher-order RRBIs even at a young age. Results showed that parents were unsure how to respond once children began their repetitive behavior. Parents showed difficulty in behavior interruption and had minimal success in re-engagement of the child in a more appropriate interaction once the repetitive behavior was allowed to occur, signifying that disruption can be difficult for parents of young children. Findings also identified that while some parents have instincts to interrupt RRBIs, their responding was inconsistent and may even inadvertently undermine the parent–child relationship (e.g., toy removal) rather than support an interactive opportunity.

### 1.3. ABA Intervention Strategies for Higher-Order RRBIs

The science of ABA examines changing socially significant behaviors by using behavior analytic principles and technologies [38]. ABA strategies in early intervention have led to developmental progress in various domains and across disability categories, especially for children diagnosed with autism [26]. ABA technologies specific to this population to address RRBIs have been identified in the research as approaches to increase flexibility during play: modeling, prompting, differential reinforcement, and response interruption and redirection. Modeling is when someone demonstrates how to perform a skill or behavior to the learner [39]. When modeling a play behavior, the parent shows how to use a toy in a new way through demonstration. Modeling is an evidence-based practice (EBP) established for increasing social engagement for toddler-aged children with ASD. Yet, it is not yet a practice with established evidence to increase play behaviors or decrease challenging behaviors for this young age group [26].

A prompt includes more direct language, and in this context, accompanied by physical support for completing a task or play expansion. Prompting is intended to reduce low-frequency or incorrect responding and is a commonly used behavioral technique to change behavior [40]. Prompting is an EBP for increasing play and social behaviors in toddler-aged children, but it is not yet established as effective in decreasing challenging behaviors for this aged child [26].

Differential reinforcement of appropriate behaviors (DRA) can increase the occurrence of the newly taught or more desired behavior while reducing undesirable behavior [41,42]. DRA is a common component in intervention packages yet has minimal evidence as a standalone intervention, therefore, suggesting its use in combination with modeling and prompting to increase intervention efficacy [43]. DRA is an EBP to decrease challenging behavior for toddler-aged children with ASD, yet it is not yet an established practice for increasing play and social engagement for this age group.

Response interruption and redirection (RIRD) is a way to eliminate or decrease behaviors that interfere or compete with other more productive or appropriate behaviors [44]. RIRD is most common in the treatment of stereotypic behavior. These behaviors can compete so intensely with other behaviors that a person cannot participate in an activity because they become too distracted by their intense interest. These restrictive behaviors, often maintained through automatic reinforcement, compete with many intervention strategies used to replace restrictive behaviors. Identifying reinforcement equivalence (i.e., a reinforcer that is as strongly desired as self-stimulation) is often extremely difficult [29]. The approach and dosage of intervention should vary depending on the type and intensity of stereotypic behavior. Yet, the intervention with the most success in decreasing the interference of restrictive, repetitive, or stereotypic behaviors is RIRD [45]. These repetitive behaviors can be quickly interrupted or can occur in tandem with prompting appropriate, alternate actions for some individuals. The use of RIRD to increase social and play behaviors while decreasing inflexibility is not yet an established practice for toddler-aged children. It is an EBP for preschool-aged children and beyond [26].

While some ABA strategies are deemed evidence-based for toddlers with ASD, very few strategies are established for toddler-aged children concerning higher-order RRBIs. More research on the impact of a parent-mediated intervention on higher-order RRBIs is necessary to determine efficacy, feasibility, and variability in delivering these interventions for very young children.

### 1.4. Vital Components for Accessible and Effective Early Intervention

Broadly, telehealth uses digital information and communication technologies to support the delivery of health-related education and services [46]. The most typical telehealth use in ASD service delivery has included two-way video and audio conferencing. Often, it uses consumer-ready technology (e.g., smartphone, computer, or Internet-enabled tablet) to communicate, coach, or teach intervention techniques virtually while seeing and hearing the other person [47]. Coaching via telehealth is the interaction between an early interventionist (i.e., coach) and parent using technology to communicate [48]. This method of intervention delivery from a distance can address shortages of professionals due to client geographical location, the cost-effectiveness of provider or family travel, and health concerns requiring quarantine and social distancing [48]. Additional barriers, such as time-intensive interventions and stressful over-scheduled family life, often prevent a parent from attempting face-to-face intervention to acquire the necessary skills needed to improve their child’s developmental outcomes adequately. Research findings demonstrate that when parents use technology and telehealth to further language and play skills with their young child with ASD, they report significant developmental gains and positivity toward using this delivery mode [49,50]. Further, literature reviews [49,51] have summarized that parent coaching via telehealth to assess and intervene in their child’s challenging behavior results in fruitful outcomes of decreasing the unwanted behavior [48,49,50,51,52,53,54,55,56]. Studies have shown the technology to be user-friendly, successful in coaching parents to conduct in-home interventions and maintaining parent-led interventions over time [50,51,52,53].

For intervention strategies to be useful for children with ASD, parents need to know how to promote their child’s active engagement in daily routines and activities; such interventions are called parent-mediated interventions. Barriers (e.g., inflexible behaviors, limited responsivity, and interference of highly preferred items) can delay or prevent play skills and social connections, even with a child’s familiarity with their primary caregivers. Unaddressed in intervention, these inflexible child behaviors can lead to isolating play and unresponsive parenting due to a child’s desire to be left alone, resulting in potential detached emotional connections, parental stress, and low self-efficacy for parents [57,58]. Including parents in choosing goals to address these isolated and inappropriate behaviors provides the opportunity for parent empowerment and individualized interventions for the family’s social and cultural context [59].

Therapist engagement is a critical component of telehealth, making it different from self-directed computer-based resources. A pilot project by Ingersoll and colleagues [60] compared the success of parent-mediated interventions for two groups: one with therapist assistance via telehealth and the other who participated in self-directed computer modules. All child participants made developmental gains. However, those children in the group with therapist assistance for parents via telehealth had greater improvements in language and social skills compared to the self-directed computer module group. Interaction with a therapist, even though telehealth, provides significant benefits to a child’s development and parental success when using parent-mediated interventions [60].

### 1.5. Purpose of this Study

At this time, there is limited research addressing the use of parent-mediated early intervention via telehealth for toddlers at risk for ASD. Additionally, higher-order RRBIs interventions that focus specifically on the flexibility of behaviors during play are even less often studied [20]. Established EBPs (e.g., ABA strategies) promoting behavior change for these young children specific to social engagement, play skills, and a reduction in challenging behavior are minimal. We aimed to examine the effects of a parent-mediated intervention as means to (a) increase child flexibility, (b) decrease child inflexibility (i.e., higher-order RRBIs), and (c) increase parent’s use of ABA strategies during play. This single-case experimental design (SCED) study targeted inflexibility in toddlers at risk for ASD. Parent–child dyads were paired with the researcher who served as the early interventionist via telehealth. The researcher coached parents to implement a treatment package of evidence-based strategies with their child during parent–child play. We hypothesized that coaching parents to use and thoughtfully sequence these strategies would result in the desired change in their child’s flexibility during parent–child play. The following research questions were tested: (1) Is there a functional relation between the parent education and coaching intervention package and the number of strategies used by the parent? (2) Is there a functional relation between the parent-implemented intervention package and the child’s flexible and inflexible behavior during play? Parent-reported ASD symptomology, including RRBIs, were measured pre-and post-intervention. Parents also reported their perceptions of the intervention package and delivery mode during and following the intervention.

## 2. Participants, Materials, and Methods

The study was conducted according to the guidelines of the Declaration of Helsinki and approved by the Institutional Review Board of the University of Oregon (protocol #08012019.001 and 14 January 2020). Informed consent was obtained from all subjects involved in the study.

### 2.1. Participants

Six children and their mothers participated in the study. Qualifying children (a) were between 18 and 35 months old at the time of consent, (b) demonstrated challenging levels of RRBIs, as reported by parents during intake assessments (Repetitive Behavior Scale—Early Childhood; RBS-EC [61] scores of 2 or higher, designating that the behavior occurs several times a day) and (c) were on ASD diagnostic or eligibility waitlists. Parents were all mothers and (a) had guardianship or legal decision-making powers for the participating child, (b) lived in the same household as the participating child, and (c) had access to Wi-Fi Internet and an Internet-capable device with Bluetooth™ connectivity. Participating families were (a) under-served if their child received equal to or less than two hours of one-on-one or group non-ASD specific EI per week or attended child care or preschool that was not specific to EI for ASD, (b) lived in a rural community if they lived in a geographic area that is at least 30 miles by road from an urban community and contained fewer than 50,000 people, [62] or (c) under-resourced if the parent reported *not having enough money* and being *never or rarely able to buy nice things* as specified on the intake form. Table 1 summarizes child participant characteristics, including sex, diagnosis, and child age in months at recruitment. The focus behaviors listed were collaboratively selected by the researcher and parent based on the RBS-EC with scores of either three or four at pre-assessment, suggesting potential intervention targets. Table 2 describes the parent characteristics.

### 2.2. Setting

All assessment and experimental sessions occurred through telehealth with the parent and child in their home in either Oregon (distance from participant to researcher’soffice *M* = 60 miles, range 15–140 miles) or Texas (distance from participant to researcher’s office was 2100 miles) and the researcher initiated sessions from a private office in Oregon. Parents selected a room in their home where they could comfortably play on the floor with their child, minimize interruptions from other family members and pets, and manage the intervention materials.

### 2.3. Researcher Roles

The first author, a white, female, board-certified behavior analyst with 25 years of experience, served as both the lead researcher and early interventionist for each family. Two trained Special Education doctoral students coded randomly selected videos of experimental sessions for procedural fidelity and dependent variables. Before the study, data collectors reached at least 90% agreement with the first author using practice videos.

### 2.4. Telehealth Equipment

The researcher used a university-issued 13-inch Dell™ laptop with an internal video camera and speakers, 2.7 GHz Intel Core i5, and 8 GB of DDR3 memory. The parent used their tablet, laptop, or smartphone equipped with a web camera and an internal speaker. Each device was connected to the family’s wireless, password-protected Internet network, and encrypted audio-visual communication was achieved through commercial videoconferencing software. A Yamay M98 Bluetooth™ headset was provided to each parent for bidirectional communication with the researcher. The researcher trained parents to use the equipment during a phone meeting before baseline data collection. A tripod was provided to parents for either their smartphone or tablet as needed.

Telehealth sessions were completed using either VSee or Zoom™ video conferencing software. Both secure audio-visual communications use 128-bit encryption that meets the requirements of the Health Insurance Portability and Accountability Act of 1996 (HIPPA), Pub. L. 104–191, 42 USC. §§ 1320 d et seq. Sessions were recorded for data collection using ApowerREC™, a computer installed screen recorder. The cloud-based and HIPPA compliant Box™ stored recorded materials.

### 2.5. Intervention Materials

Each participating family was mailed a tote bag filled with toys to use during baseline and intervention sessions. Play materials consisted of toys in the *Short Play and Communication Evaluation* [63], which is an ASD-specific play assessment. The chosen toys covered differing levels of play and developmental skills. Providing items specific to each play level allowed children to perform independently based on their development. Some toys included multiple pieces (e.g., set of Duplo™ blocks, wooden inset-puzzle). If pieces got lost during the study, the toy was still usable, preventing compromised fidelity due to materials loss. The play materials cost $125.68 US dollars per child. Families kept the toys after the study, providing an additional incentive. The families were encouraged to use the new toys combined with toys the child already had. The interventionist encouraged parents to allow their child to access the bag of toys anytime but with parent participation as often as possible.

Educational materials and instructions were provided in a parent binder. Materials in the binder were labeled “pre-baseline”, “baseline”, and “intervention”, designating when they were to be accessed by the parent. Materials that could not be viewed by the parent before particular time points in the study (e.g., intervention strategies at baseline) were marked and clipped with a binder clip. Pre-baseline materials included (a) consent and assent information, (b) contact information for the interventionist, (c) telehealth instructions, and (d) a list of toys included in the tote. Baseline materials included (a) written instructions for sessions, (b) worksheets to identify target behaviors, and (c) replacement behavior worksheets. Intervention materials included (a) four strategy overviews, (b) instructions for strategy videos, (c) ideas for play activities to engage their child, and (d) instructional charts detailing the coaching format and session timeline.

The researcher developed three videos demonstrating intervention strategies to introduce each concept to parents: (a) video 1: modeling and prompting; (b) video 2: differential reinforcement of appropriate behaviors; and (d) video 3: response interruption and redirection. These videos were made available on Box™ for each family at the onset of the intervention phase. Once a strategy was introduced and the video was presented to the parent, it remained available for repeated viewing.

### 2.6. General Procedures

All experimental sessions occurred via telehealth, and the family was asked to have the provided toys, parent binder, and technology available. This study included the four phases: (1) intake, (2) baseline, (3) intervention, and (4) post-intervention.

#### 2.6.1. Phase 1: Intake

Assessments given at intake included the RBS-EC [61] and the Modified Checklist for Autism in Toddlers, which were revised with follow-up (M-CHAT-R/F) [64].

The RBS-EC [61] evaluated a child’s RRBIs. This scale took approximately ten minutes to complete and is a standardized measure to more accurately discriminate between RRBs, which are considered typical for the developmental age versus those which act more as barriers to development [61]. This scale can be used for children from toddlerhood to early school-age. It asks the parent to circle the frequency of behavior on a Likert-type scale between 0 (does not occur) and 4 (occurs many times per day). The scores from this measure were disclosed to the parents to determine intervention focus. This measure was given at the start and completion of the study.

The M-CHA-T-R/F [64] was a screening tool for children between the ages of 16 and 30 months that requests the parent complete a short assessment form to identify child behavior, and signifies if they are at risk for autism. This pre- and post-measure assisted in understanding unique intervention needs. The scores from this measure were not disclosed to the parents.

Questionnaires included general demographic information (intake form available on request from authors). Intake was completed in a 45-min session between parent and PI.

#### 2.6.2. Phase 2: Baseline

Concurrent observations established baseline performance for each dyad (within one day of each other). During baseline probes, parents were asked to play with their child as they usually would using the researcher provided toys and toys of their own. The researcher did not provide any feedback to the parent during baseline. Baseline duration was between 5 and 11 sessions based on the dyad’s tier location in the multiple-baseline design and parent performance. During the baseline phase, each parent and child dyad had two meetings per week with the researcher. Each meeting lasted 10 min total with only one baseline data point per meeting. Following the baseline data collection, a post-baseline session occurred between the parent and PI to identify and operationally define inflexible behaviors to decrease and flexible behaviors to use as replacement behaviors using the completed child assessments. Inflexible behaviors were identified as interests that occur more frequently than others and interfere with social opportunities. Inflexible behaviors to decrease during the intervention were individual to each participant and included negative responses (e.g., whining, screaming, hitting) toward parent’s attempt to interact; repetitive movements (e.g., flipping, mouthing, throwing toys) or interests (e.g., plays with only one toy) were responses that interfered with or prevented interactions or expansions of play. Flexible behaviors to use as replacements for inflexibility were behaviors that the child previously exhibited, rendering increased flexible and responsive interactions between parent and child. Flexible behaviors included exchanging materials and toys, verbally or physically acknowledging the parent during play, and changing play focus to new ideas introduced in the environment or parent. An overview of the intervention session sequence (Table 3) and the bi-directional coaching components [65] (Table 4) was also discussed.

#### 2.6.3. Phase 3: Intervention

The intervention phase began with an education session (i.e., session 1). The researcher presented the parent with the rationale and procedures for the four intervention strategies (25 min) followed by an initial coaching session (20 min). The researcher provided immediate feedback and error correction contingent on parent and child responses.

After this initial education session, each session followed the sequence prescribed in Table 3. For the remaining intervention sessions (sessions 2–15), each dyad met with the researcher twice sessions per week. Each session lasted 40 min, resulting in 80 min of intervention per week. During each intervention session, a 10-min data probe of parent–child play without coaching (labeled *dyad interaction* in Table 3) was conducted to track behavior change and collect data implementing baseline conditions. Coaching was provided after the 10-min dyad interaction was completed (labeled *parent coaching* in Table 3). While no feedback or guidance was given during the dyad interaction segment of the intervention session, feedback was given during the parent coaching segment of each intervention session. The researcher provided verbal guidance and feedback during coaching throughout each strategy sequence as necessary for each parent’s individual comfort level and child performance.

Four behavioral analytic EBPs specifically targeting inflexible behavior were taught to parents in the context of parent–child play [26,43,66,67]. The following EBPs were used and taught to each parent: (1) modeling [39,40,67], (2) prompting [67,68], (3) differential reinforcement of alternative behaviors [41,42], and (4) RIRD [44,45].

For teaching flexibility during play, modeling (i.e., parent-enacted actions with materials and turn-taking) was used first. If the child did not respond to this initial demonstration of flexibility, a second model was presented. Research supports the use of modeling when teaching imitation, including the combination of modeling and time delay (pre-prompt interval of 5 s allowing for child response) [69]. Adding in a time delay after a model allowed the child time to respond, especially if receptive or expressive skills are underdeveloped

If the model did not elicit a flexible response after this second attempt, the parent used a prompt (i.e., hand over hand physical encouragement, suggestive pointing, and child positioning). The parent added a time delay after their prompt to wait and see if their child continued with the suggested play (e.g., imitated action, extension, request for more interaction). Since this required more direction from the parent than a mere example as in modeling, prompting was considered more intrusive than modeling in this play context. Chaining these strategies (i.e., modeling then prompting) provided graduated guidance for the child.

Differential reinforcement of the child’s appropriate behavior included reinforcement specific to the individual dyad (e.g., preferred type of reinforcement by the child and preferred mode of delivery by the parent). Tangible reinforcers were not introduced to keep the play interactions as natural as possible. The parents’ reinforcement strategies included verbal praise, physical touch, proximity, attention, and extensions in the play exchanges.

RIRD was the most directive intervention strategy introduced to parents. This strategy targeted restrictive and repetitive behaviors that interfere with a child’s ability to change focus to what the parent was modeling and prompting as flexibility during play. Interruption of inflexible behavior (e.g., overly focused on repetitive play actions) was taught to be minimally intrusive and practiced with the parent to decrease the likelihood of challenging behavior. For example, RIRD may have included the parent resting a hand on the child to temporarily impede further repetition or playfully tickling the child to break their focus on the action or toy inspection. The parent’s redirection may have included a different, highly reinforcing toy or model of new actions to direct the child’s play behavior to a more flexible interaction. Similar to the prompt, once the parent interrupts and redirect their child, a 5 s time delay occurred to see if the child continued the suggested action.

Strategies were introduced as a graduated sequence of strategies, always beginning with the least intrusive strategy (i.e., modeling) and moving through each strategy with graduated guidance until the child could reciprocate a flexible response to the parent. If the parent could not elicit a flexible play response from their child, the intervention sequence was stopped. The PI and parent worked to adjust the intervention strategy to meet the child’s individual needs more effectively. Samples of individual dyad lesson plans are available by request from the first author.

Based on clinical judgment, a rate of two strategies per minute was identified as an ample number of interactive models and prompted during the 10-min play session. Thus, there was a ceiling of 20 sequences per play session. Details of the Phase 3 intervention sessions that involved coaching are provided in Table 3. Fifteen coaching sessions were offered to each parent as part of the intervention phase across 12 weeks. Dyad 1 was lost to attrition before intervention due to COVID-19 pandemic issues. Figure 1 demonstrates the sequence of strategies using the least to most intrusive steps of intervention.

#### 2.6.4. Phase 4: Post-Intervention

In the final phase four, the research held an optional post-intervention follow-up meeting with the family and their other providers. Parents were asked to complete the post-intervention assessments.

### 2.7. Experimental Design, Analysis, and Response Measurement

#### 2.7.1. Experimental Design and Analysis

An SCED was used to examine the effects of the coaching via telehealth on parent use of targeted intervention strategies and parent implementation of the intervention package on targeted child outcomes. SCEDs are causal designs that differ from qualitative case designs using repeated outcome measures, using baseline measurement to document an issue of social concern before the intervention. Additionally, the selection of designs allows for demonstrating a basic intervention effect across, at minimum, three different time points [70]. A concurrent multiple-baseline design across six parent–child dyads with baseline, intervention, and follow-up phases was planned to allow for six opportunities to demonstrate a basic effect when the intervention was introduced. A multiple-baseline design staggers the introduction of intervention in a time-lagged fashion to assess behavioral covariation across the participants during an overlapping period of time; an increasing baseline trend would suggest a threat to internal validity due to variables extraneous to the intervention. As such, a multiple-baseline design requires three or more participants with staggered baseline lengths of a minimum of five data points before the beginning of the intervention [70].

Visual analysis of a graphed dependent variable data and a calculation of the within and between-effect size were used to estimate the treatment impact. An analysis of the levels, trends, overlap, and variability of the dependent variables within (each A-B comparison where A is baseline and B is intervention) and across participants (tiers) of the multiple-baseline design [70] examined whether a causal relation could be inferred from the data paths. Following visual analysis, a non-overlap estimator that accounts for the baseline trend, Tau-U, was calculated to statistically estimate the amount of change following intervention for each participant [70,71]. To estimate between-case effect size, we calculated a standardized mean difference (SMD) [70,72].

#### 2.7.2. Measures for the Dependent Variables

Child behavior was measured using frequency counts per 10 s interval of inflexible behaviors and flexible, responsive behaviors (as defined for individual dyads in Phase 2 of the intervention). The dependent variable was the parent’s correct use of intervention strategy sequences, as measured by the frequency of correct steps in sequence (as described in Figure 1) during each 10-min play session at the baseline and intervention phases. Both parent and child behaviors were measured across the baseline and intervention phases during parent–child play. Dependent variable data were collected using the iPhone application Insight: Observation Timer Tool for School Psychologists.

#### 2.7.3. Pre and Post Measures

The study also used non-experimental data collection and analysis. All pre- and post-measures reported change over time, comparing raw scores between the pre-baseline and post-intervention assessments. Child pre- and post-measures helped identify target behaviors and tracking changes over time (i.e., RBS-EC [61] and M-CHAT-R/F [64]).

### 2.8. Fidelity, Reliability, and Social Validity

#### 2.8.1. Coach Fidelity

All sessions were video-recorded, and researcher fidelity was assessed using researcher-developed checklists for the first four sessions and 30% of each dyad’s remaining sessions. Coaching fidelity was above 90% in coaching procedures. The coaching fidelity checklist is available upon request.

#### 2.8.2. Parent Treatment Fidelity

Intervention strategies used by the parents were broken into a sequence of steps. Partial and complete sequences were tracked during 10-min play sessions. Then, the number of total steps completed correctly within a sequence was divided by the number of steps needed to complete the sequence and multiplied by 100 to obtain a percentage ranging from 0% to 100% of steps completed correctly. The researcher used the steps skipped or missed within a sequence to determine additional coaching support areas. Parent fidelity data were collected for 100% of intervention sessions. See Table 5 for parent treatment fidelity per dyad.

No treatment fidelity was collected for dyad 1 due to attrition prior to intervention. For dyad 2, overall, parent treatment fidelity was 95% (range = 73–100%) for 100% of intervention sessions. For dyad 3, overall parent treatment fidelity was 93% (range = 67–100%). For dyad 4, overall, parent treatment fidelity was 97% (range = 91–100%) for 100% of intervention sessions. For dyad 5, overall, parent treatment fidelity was 97% (range = 81–100%) for 100% of intervention sessions. For dyad 6, overall, parent treatment fidelity was 97% (range = 92–100%) for 100% of intervention sessions.

#### 2.8.3. Reliability

Inter-observer agreement (IOA) data for dependent variables (i.e., child behavior and parent strategy use), and parent fidelity was collected by reviewing a random selection of 30% of baseline and 30% of intervention sessions for each dyad. Point-by-point IOA was calculated by subtracting disagreements from agreements, dividing by the number of possible outcomes, and multiplying by 100 to obtain a percentage of child behavior and parent strategy use [69].

Inter-observer agreement (IOA) data were collected for 37% (range = 35–40%) of baseline sessions and 36% (range = 34–36%) of intervention sessions across dyads for child behaviors and parent strategy use. IOA data were collected for 31% (range = 29–36%) of intervention sessions for parent treatment fidelity. Research assistants were provided with behavioral definitions and trained by the PI using sample videos until they could reach a minimum of 90% agreement across three consecutive sessions before beginning formal data collection. The PI randomly selected videos for IOA data. See Table 5 for reliability of child behaviors, parent strategy use, and parent treatment fidelity.

#### 2.8.4. Intervention Procedural Fidelity

One-hundred percent of coaching sessions were coded for procedural fidelity by the PI using a fidelity checklist. During intervention sessions with dyad 2, the interventionist averaged 98% procedural fidelity (range = 91–100%). For dyad 3, the interventionist averaged 99% procedural fidelity (range = 91–100%). For dyad 4, the interventionist averaged 100% procedural fidelity. For dyad 5, the interventionist averaged 99% procedural fidelity (range = 82–100%). For dyad 6, the interventionist averaged 98% procedural fidelity (range = 91–100%).

The IOA coder used the Fidelity Checklist for Coaching Sessions to assess the continuity across a comprehensive set of topics and telehealth procedures for 33% of sessions. IOA was calculated using the point-by-point method described above. IOA for intervention procedural fidelity is included in Table 5.

#### 2.8.5. Social Validity

Using an adapted version of the *Treatment Acceptability Rating Form-Revised* (TARF-R) [73], parents were asked to complete a social validity questionnaire mid- and post-intervention to assess the usability, compatibility, and efficacy of the intervention package components. Participants completed an evaluation of the social validity of the telehealth procedures that used a version of the TARF-R [73] specific to telehealth delivery.

## 3. Results

### 3.1. Descriptive Results

Minimal data were collected for dyad 1 due to attrition post-baseline. For all other dyads, descriptive results are presented. Vertical analysis for dyads 1–6 shows that each tier’s behavior change is independent of the other tiers. Figure 2 shows the tiered, graphed data for each dyad’s baseline and intervention.

#### 3.1.1. Dyad 1: Bree and Jax

Bree’s use of behavior strategies was 0 for the five baseline sessions she completed. During baseline, Jax’s inflexible and flexible behaviors were variable. Flexible behaviors averaged 42% of the intervals, and inflexible behaviors averaged 46% of the intervals. No intervention data were gathered due to attrition.

#### 3.1.2. Dyad 2: Vicki and Maude

Vicki’s use of behavior strategies was 0 for all five baseline sessions, demonstrating stable responses. Vicki’s responses remained at zero for the first data point in the intervention phase. During the next sessions, Vicki exhibited an increasing trend in strategy use. Vicki used an average of 11 strategy sequences per 10-min play session, increasing from the average of 10 initially to 13 in the last four sessions. Vicki’s overall average rate of sequence use during intervention was 1.0 per minute. Vicki used the model 1 sequence most frequently (71% of her strategy sequences), with the prompt sequence next in frequency (23% of her sequences). The most frequently missed sequence component was DRA, at 77% of missed components (see Table 6). Tau-U analysis showed a strong basic effect (*B-A difference* = 0.93; A = baseline and B = intervention) on parent strategy use.

During baseline, Maude demonstrated stable, low levels of flexible behaviors at an average of 16% of an interval and stable high levels of inflexible behaviors, averaging 62% of an interval. During the intervention, Maude initially demonstrated low flexible behaviors with only a slight decrease in her inflexible behavior. During the next sessions, Maude showed an increasing trend in flexibility, averaging 68% of intervals, and decreased inflexible behaviors, averaging 14% of the intervention intervals.

All behaviors showed a change in level and minimal to no overlap between baseline and intervention. A vertical analysis demonstrated behavior change for Tier 2 to be independent of the other tiers. Tau-U analysis showed a strong basic effect on flexible (*B-A difference* = 0.99) and inflexible behaviors (*B-A difference* = 1.00). No data were collected for dyad 2 for session three due to a break between intervention sessions 1 and 2 and intervention sessions 2 and 3.

#### 3.1.3. Dyad 3: Liz and Daisy

Liz’s behavior strategies were 0 for all seven baseline sessions, demonstrating stable responding. During intervention, Liz’s began with one sequence and increased from there, demonstrating an upward trend. Liz used an average of seven strategy sequences per 10-min play session, increasing from an average of four sequences in sessions 2–10 up to 15 sequences in the last sessions. Liz’s overall average rate of sequence use during intervention was 1.3 per minute. Liz used the model 1 sequence most frequently (84% of her strategy sequences), with the prompt sequence next in frequency (8% of the strategies sequences used). The most frequently missed sequence component was DRA, making up 91% of missed components (see Table 6). Tau-U analysis showed a strong basic effect (*B-A difference* = 1.00) on parent strategy use.

During baseline, Daisy demonstrated stable, mid-levels of flexible behaviors with an average of 58% of the intervals and stable low-levels of inflexible behaviors at 17%. During intervention, Daisy’s responding remained similar during the first few intervention sessions; however, once Liz increased her number of strategy sequences, Daisy also increased her flexible behaviors, averaging 67% of intervals, following the same upward trend with variability as her mother. Alternatively, Daisy’s inflexible behavior remained low (averaging 5% of intervals) and decreased further to zero and near-zero responding. As Daisy’s flexible behavior began at a higher level in baseline, Tau-U results indicated a weak basic effect (*B-A difference* = 0.65) for flexible behavior and a strong basic effect (*B-A difference* = 0.98) on inflexible behavior.

#### 3.1.4. Dyad 4: Gigi and Lucia

Gigi’s use of behavior strategies averaged 0.9 (range 0–3) for the seven baseline sessions, demonstrating minimal variability. During intervention, Gigi’s responding showed an immediacy of change, with a change in level and increasing trend. In sessions 2–10, Gigi averaged nine strategy sequences, and in sessions 11–15, she averaged 14 sequences for each play session. Gigi used an average of 10 strategy sequences per 10-min play session. Gigi’s overall average rate of sequence use during intervention was 1.03 per minute. Gigi used the model 1 sequence most frequently (88% of her strategy sequences) with the prompt sequence next infrequently (7% of her sequences). The most frequently missed sequence component was DRA, at 67% of missed components. Gigi was the only participant to use the RIRD sequence, which she used once (see Table 6). Tau-U analyses showed a strong basic effect (*B-A difference* = 1.00) on parent strategy use.

Lucia demonstrated stable, low levels of flexible behaviors (average 20%) and stable high levels of inflexible behaviors (average 54%) during baseline intervals. During intervention, Lucia’s flexible behavior showed minimal overlap with baseline for the first data point. Then, it began an increasing trend without further overlap and after that a change in level and trend. Flexible behavior increased (average 61%), and inflexible behaviors decreased (average of 29%) across intervention. Tau-U analysis showed a strong basic effect for child flexible (*B-A difference =* 0.99) and child inflexible (*B-A difference* = 0.91) behavior.

#### 3.1.5. Dyad 5: Kay and Derek

Kay’s use of behavior strategies averaged 0.6 (range 0–3) for the nine baseline sessions demonstrating minimal variability. During intervention, Kay’s responding overlapped with baseline levels for the first two data points, which was followed by an increasing trend, change in level, and minimal overlap. In sessions 2–10, Kay averaged eight strategy sequences, and in sessions 11–15, she averaged 17 sequences. Kay used an average of 12 strategy sequences per 10-min play session. Kay’s overall average rate of sequence use during intervention was 1.2 per minute. Kay used the model 1 sequence most frequently (95% of her strategy sequences) with the prompt sequence next in frequency (4% of her strategy sequences). The only sequence component missed by Kay was DRA (see Table 6). Tau-U analyses showed a strong basic effect (*B-A difference* = 0.95) for parent strategy use.

Derek demonstrated mid-range responding, with high variability for flexible behaviors (average 43%) and inflexible behaviors (average 35%) for baseline intervals. During intervention, Derek’s behavior showed minimal to no overlap with a change in level and trend; flexible behavior increased (average 79%), and inflexible behavior decreased (average 7%). Tau-U analyses showed a strong basic effect for child flexible (*B-A difference* = 0.95) and inflexible (*B-A difference* = 0.96) behavior.

#### 3.1.6. Dyad 6: Maria and Allie

Maria used very few strategy sequences during baseline, with an average of 0.3 sequences across 11 baseline sessions demonstrating minimal variability. During intervention, Maria’s responding remained low for the first three data points and then began an increasing trend and change in level. In sessions 2–10, Maria averaged eight strategy sequences, and in sessions 11–15, she averaged 16 sequences. Maria used an average of 11 strategy sequences per 10-min play session. Maria’s overall average rate of sequence use during intervention was 1.1 per minute. Maria used model 1 sequence most frequently (56% of her sequences) overall with the prompt sequence (35% of the strategies used). Maria’s most frequently missed sequence component was DRA at 69% of missed components (see Table 6). Tau-U analyses showed a strong basic effect (*B-A difference* = 0.94) for parent strategy use.

Allie demonstrated variable yet low levels of flexible (average 19%) and high levels of inflexible (average 54%) behaviors during baseline. During intervention, Allie’s data points overlapped during the initial sessions then began to increase, demonstrating a change in level and an increasing trend for flexible (average 59%) and inflexible (average 21%) behaviors. Tau-U analyses showed a moderate basic effect for child flexible (*B-A difference* = 0.79) and a strong basic effect for inflexible (*B-A difference* = 0.90) behavior.

### 3.2. Standard Mean Difference (Between-Case Effect Size)

The between-case effect size shown as standard mean difference (SMD) for each dependent variable was above zero, and for child inflexible behavior, it was below zero, as anticipated for a decrease in behavior. For parent strategy use, SMD = 0.06, (95% CI [−0.006, 1.21], *df* = 16.09, *SE* = 0.29, *p* < 0.02). For child flexible behavior, SMD = 0.27, (95% CI [−0.28, 0.084], *df* = 15.73, *SE* = 0.26, *p* < 0.20). For child inflexible behavior, SMD = −1.21, (95% CI [−2.11, −0.29], *df* = 9.37, *SE* = 0.40, *p* < 0.01). Thus, the probability of participants demonstrating the observed behavior change is not likely without intervention.

### 3.3. Non-Experimental Results

Parents were asked to complete two sets of child assessments, one before the start of baseline (i.e., pre-assessment) and the second after the last intervention session (i.e., post-assessment). Table 7 presents the scores for pre- and post-intervention assessments for child ASD symptomology change overtime. Changes in RBS-EC [61] scores varied across participants with some increasing, others decreasing and some staying the same. M-CHAT-R/F [64] scores decreased for all participants except Daisy, who increased between pre- and post-assessment scoring.

### 3.4. Social Validity

Adapted Treatment Acceptability Rating Form-Revised (TARF-R) [73] assessments were conducted at midpoint for two participants and post-intervention for all participants. Scores of treatment acceptability for midpoint averaged 4.4 (range = 3.8–5.0) and at post-intervention averaged 4.8 (range = 4.0–5.0). Scores on effectiveness for midpoint averaged 4.5 (range = 4.0–5.0) and at post-intervention averaged 4.8 (range = 4.0–5.0). Scores on disadvantages for midpoint averaged 1.5 (range = 1.0–2.0) and at post-intervention averaged 1.6 (range = 1.0–3.0). Scores on contextual fit was 5.0 at midpoint and at post-intervention averaged 4.8 (range = 4.0–5.0).

Additionally, social validity assessments specific to telehealth procedures were conducted post-intervention. Scores for treatment acceptability averaged 4.4 (range = 4.0–4.8). Scores on effectiveness averaged 4.7 (range 4.0–5.0). Scores on disadvantages averaged 1.9 (range 1.0–3.5). Scores for contextual fit averaged 4.5 (range = 4.0–4.6).

## 4. Discussion

This SCED study evaluated the impact of a parent-mediated intervention package delivered via telehealth coaching on child flexibility for six children on a waitlist for an ASD diagnosis. This study had two primary aims. First, it aimed to determine if a functional relation existed between implementing the parent education and coaching package and increased frequency of targeted parent strategy use and if strategies were perceived as acceptable and feasible to parents. The second aim was to determine if the parent-implemented intervention package impacted child inflexibility (i.e., higher-order RRBIs) and flexibility during parent–child play and the magnitude of treatment effects. The findings, limitations, and implications for research are discussed.

### 4.1. Parent Strategy Use

Each of the five parents increased their targeted intervention strategies during the intervention phase. For the five parents completing intervention, each demonstrated an immediate change in their use of the strategies within the first three intervention sessions. All parent participants demonstrated a strong basic effect between baseline and intervention in strategy use frequency. The most frequently used strategy sequence was model 1 sequence. The most challenging strategy sequence step (i.e., the one most frequently missed by parents) was contingent reinforcement following the child’s flexible behavior. All parent participants used 13 (range = 13–20) or more strategy sequences in the last two sessions showing an increase over time.

As recommended by past research, this intervention used *the least* intrusive model first (i.e., model 1 sequence) and graduated to the *most* guidance (i.e., RIRD sequence), based on the child’s behavioral needs [66,74]. As data collection occurred during playtime with a parent, most of the interactions between child and parent should have been least intrusive, maintaining a supportive and positive play environment [75]. If this intervention was conducted during routines that required more parent directives, such as toileting or clean-up, more intrusive prompts might have been necessary for task completion and thus used more often.

Reinforcement was the most commonly missed step by the parent participants during intervention phases. Once the interventionist used coaching to address this strategy, most participants learned to use social or object reinforcement, resulting in a decrease in missed steps as intervention progressed. However, the severity of child delay and high levels of parent stress may have impacted the use of positive parenting techniques (e.g., reinforcement) [76]. For Liz and Daisy, reinforcement was missed more than twice as many times as other participants during the intervention phase. Daisy was the only participant with multiple diagnoses at the start of the study, where the other child participants had a developmental delay. Additionally, during the study, Daisy had two hospitalizations (one for the flu and another for seizure activity); Liz also underwent surgery herself, endured a sprained ankle, and changed jobs twice. It is possible that due to Liz’s life stressors combined with Daisy’s multiple diagnoses, reinforcement (e.g., positive praise) was challenging to maintain.

### 4.2. Child Flexible and Inflexible Behavior

All child participants demonstrated a decrease in inflexible behavior and an increase in flexible behavior during the intervention package implementation. For four of the five participants, a strong basic effect was identified between the use of the intervention package and increased flexible behavior during play. Daisy was the exception, demonstrating a weak basic effect between the intervention and her flexible behavior during play. Daisy demonstrated the highest levels of flexible play compared to the other children in the study. Although her flexible behavior demonstrated an increasing trend, it was highly variable and frequently overlapped with her baseline performance.

A parent with strong play arrangement skills could increase flexibility with the least intrusive sequence. The dyad with the highest play engagement levels was Derek and Kay, which likely had to do with Kay’s background as a preschool teacher. Despite experience in playing together, data analysis demonstrated a strong basic effect of the intervention on Kay and Derek’s behavior. Kay’s initial assessment of Derek’s higher-order RRBIs was in the problematic range of 3 s and 4 s on the RBS-EC; however, Derek responded to Kay’s change in interaction. For example, during intervention session 8, both dyad participants had high responses: Kay used sixteen sequences; Derek used flexible behavior in 72% of the intervals and inflexible behaviors in only 5%.

Research has demonstrated that RRBIs frequently occur for toddler-aged children with ASD during play, and researchers have explored some areas of play diversity in clinical and classroom settings [77]. For example, Frey and Kaiser [66] measured object play diversity by modeling imitation. Hanley and colleagues [78] used embedded reinforcement to influence preschool-aged children’s classroom activity preference. Several studies have reported lag schedules of reinforcement to increase the variety of play in classroom settings [79,80,81]. Various studies investigating the use of parent–child interaction therapy demonstrated a decrease in child problem behavior during play [82,83]. Yet, measuring play diversity under natural contingencies is less frequently studied in behavior analytic research. Using natural play environments (i.e., in the home or community settings) allows for identifying child and parent characteristics and active ingredients necessary to promote play diversity in toddlers at risk for ASD. Future research should include the measurement of flexible and inflexible behaviors and play diversity as supported by behavior analytic strategy sequences used in this study.

### 4.3. Parent Strategy Use as a Moderator for Child Flexibility

Both parent strategy-use and flexible child behavior during play increased, while inflexible child behavior decreased. On average, all parent participants used the highest cumulative number of strategy sequences during the last five sessions of the intervention condition. Similarly, on average, all child participants demonstrated their highest levels of flexible behaviors and lowest inflexible behaviors during the last five play sessions of the intervention condition. Although the intervention protocol outlined parent strategy use for individual RRBIs, the protocol for strategy use remained the same for all participants: start with the least intrusive prompt and graduate to the most directive prompt. At intervention session six, Vicki shared, “*I never thought I’d see so many changes so fast*,” when Maude imitated several behaviors that Vicki demonstrated in the session. Maude continued to progress as Vicki used strategies; however, there were incidences where Vicki used a prompt sequence, which introduced the prompt more frequently when a model may have previously been enough support to promote her child’s flexibility. This observation led to instructions to *avoid prompt dependency* and revisit the importance of starting with a model. Coaching on how to avoid prompt dependency and fading procedures for parent strategy use should be included in the intervention package for future research.

### 4.4. Parent-Reported Autism Symptomology and Diagnosis

All participants scored within the moderate to the severe range for ASD on the M-CHAT-R/F [64] at pre-intervention. Vicki’s rating of Maude’s post-intervention behavior decreased on the M-CHAT-R/F despite Maude having received an autism diagnosis. Liz increased her score for Daisy in post-intervention despite Daisy receiving a preliminary/inconclusive diagnosis of ASD during her diagnostic visit. Past research has shown that the M-CHAT-R/F is not always accurate in identifying ASD [84]. Thus, such measures should be cautioned when drawing conclusions as parental concerns may not be reflected by screener’s alone.

While Maude (dyad 2) and Daisy (dyad 3) did receive virtual diagnostic visits that resulted in ASD diagnosis, the remaining dyads had different experiences. Lucia (dyad 4) received an ASD diagnosis six-months post study. Derek (dyad 5) was removed from the waitlist by his parent. Allie (dyad 6) received an ASD diagnosis within one month of the intervention conclusion. The length of time each child was on a wait list varied by state and county. Wait times ranged from 5 to 20 months. Considering the current sequence of *diagnosis first, intervention second*, access to intervention was delayed considerably for some participants.

All participants completing the intervention demonstrated a decrease in the frequency of their inflexible behavior during play sessions with a strong basic effect (i.e., Tau-U). Participants had similar ratings on the RBS-EC [61] used to identify individual behaviors of focus and create contextually sound intervention plans based on these behaviors. The RBS-EC was used as a pre-and post-measure, resulting in some change at the two time points. Parents likely reported on their child’s behavior overall, throughout the day, and during various routines. This study merely reported the child’s behaviors during observed parent–child play. Therefore, it is essential to highlight that this study’s data demonstrates changes in child flexibility only during parent–child play. Simultaneously, the parent’s assessments may represent a more comprehensive picture of the child’s behavior. Future research is recommended to consider the routines in which most inflexible behavior occurs and build coaching sessions around these routines to target the highest areas of need for the family.

### 4.5. Limitations

Limitations of this study do exist. First, the intervention duration was brief (fifteen sessions per dyad) and was insufficient to ensure equal opportunity for all strategies to be tested (i.e., RIRD). Second, while a multiple baseline design controls for the influence of many external influences, the rigor of this SCD could have been improved with randomization of strategy use and introduction among the participants. Third, generalization and maintenance were not programmed within this intervention. Using the strategies to attend to other routines and behaviors were discussed; however, these additional routines were not observed or measured. Additionally, if generalization and maintenance is used in future research, measurement should rely on data gathered in session rather than by a general measurement tool (e.g., MCHAT-R/F). As stated previously, such measures are not sensitive enough nor designed to measure behavior change as demonstrated by intervention specifically, Fourth, the intervention was designed to target inflexible and flexible behaviors during playtime, yet skills specific to play were not taught to the child or the parent. Additionally, other behaviors common for children with ASD (e.g., social reciprocity, facial affect, communication) were not measured. Finally, coaching of this parent-mediated intervention was researcher delivered. Therefore, the feasibility, acceptability, and effectiveness of coaching delivered by early intervention practitioners are unknown.

### 4.6. Implications for Research and Practice

The wait time for a diagnostic visit and ASD-specific intervention is too long for families. Virtual visits are one way to address this demand to provide services sooner. Research on the effectiveness of virtual visits is improving to ensure reliability, usability, and accuracy [85,86,87,88,89], making this a good option for families to receive services despite geographic location and service availability. Increased access to intervention could ultimately eliminate the “*diagnosis first, intervention second*” sequence, allowing children to receive necessary early intervention based on behavioral and developmental characteristics rather than a diagnostic label.

Interventions that focus on social communication in toddler-aged children with ASD are established in early intervention research and prominent in practice [22,24,90,91]. In contrast, interventions for RRBIs for this age group are less developed [37,90]. The current study addresses this gap in the literature by using intervention strategies (i.e., modeling, prompting, differential reinforcement, and RIRD) to address inflexibility during the natural context of play with their parent. While outcomes from this study were successful in parent strategy use and change in child behavior, additional research is necessary to establish evidence-based intervention for RRBIs for toddlers at risk for ASD.

All intervention strategies used in this study are identified as evidence-based practices for preschool-aged children and above, with only some established evidence for toddlers with ASD [26]. More specifically, within the category of challenging and interfering behavior, differential reinforcement, antecedent-based intervention, and more broadly naturalistic and parent-implemented interventions are considered evidence-based practices for toddlers with ASD. Modeling, prompting, and RIRD require further research to establish an evidence base with their use with these younger children. The current study does support the use of modeling and prompting; however, RIRD was used too infrequently following the brief parent training and coaching to suggest it impacted inflexibility in child participants. Therefore, future research should establish the effectiveness and appropriateness of RIRD for toddlers around challenging and interfering behavior. The next phase of intervention development should include additional components (i.e., fading, play diversity, prompt dependency, and preference assessment) to explore further the impact of ABA technologies on RRBIs with this population.

The current study addressed inflexible and flexible behaviors during play; however, RRBIs as defining characteristics of ASD are broader than the behaviors investigated here. RRBIs may be less frequent, less intrusive, or, perhaps, parents are more tolerant of these behaviors in toddlerhood [92,93]. Qualitative studies to describe parent tolerance and perception of RRBIs in toddler-aged children should be conducted. The RBS-EC was used to measure RRBIs in this study. This scale is not meant to identify intervention goals around these behaviors but rather provide a score to determine severity. The next step is developing a measurement tool and assessment, which translates RRBI identification into intervention targets to bridge this research to practice gap.

Many children who receive early intervention make developmental gains, including adaptive skills, increasing their likelihood of being part of inclusive education settings. It is unlikely that gains from this parent-mediated, naturalistic intervention could change a child’s developmental trajectory to appear *less autistic* than they did before the intervention nor was this intervention’s aim to do so. Additionally, the goal of intervention should remain focused on socially significant goals that impact quality of life not an attempt to mask autistic traits. It is essential to consider the intervention’s goal: to increase flexibility and decrease behaviors that interfere with social interactions between child and parent. While these characteristics are associated with autism symptomatology, the goal here was not to eradicate these behaviors but to change the intensity and frequency of interfering behaviors to improve one’s quality of life. Future research and practice should consider the impact of RRBIs to ensure the end goal is socially significant to the child, family, and stakeholders rather than behavior change for the sake of change alone.

### 4.7. Future Research

The results from this study revealed a strong basic effect between the researcher-led, parent-mediated intervention package, parent strategy use, and flexible and inflexible child behavior. Participants rated the intervention as highly effective and usable. Future research should include modifications of this initial intervention into a more scalable intervention with broad applicability. Specific modifications could include a broader range of children (e.g., with and at risk for ASD and varied developmental delays) to eliminate barriers of diagnosis before intervention and increase family’s access to skilled therapists and interventionists. Generalization of the intervention strategies beyond one environment (e.g., in-home play, routines, and community settings) could meet various targets identified by parents. An accessible training program for early intervention providers to coach parents in behavior analytic strategy use is recommended. Last, future research should use an advisory board and community-engagement research model to address the controversy of RRBI intervention and preserve the central voice of the child and family in early intervention.

## 5. Conclusions

This study demonstrated the utility and effectiveness of coaching via telehealth on increased parent use of intervention strategies addressing inflexible child behaviors for toddlers on the waitlist for an ASD diagnosis. Outcomes demonstrated the positive effects of this parent-implemented intervention package during play on their young child’s flexible and inflexible behaviors. The use of parent-mediated interventions to change child behavior is not new [92]; however, various components of this study are unique and contribute to future practice. First, the strategies used to influence behavior change echo previous research on modeling, prompting, time-delay, and reinforcement [29,34,67,94]. This study contributes explicitly to the potential influence these technologies have on toddler-aged children’s flexible play behavior. Past research has examined play skills as intervention targets in contrast to targeting child flexibility during play [74]. In this study, the barrier behavior (inflexibility) was the focus for behavior change. Decreasing the frequency of inflexibility may allow for more interactive behaviors to occur (e.g., matching law) [94]. Few studies have targeted higher-order RRBIs, and even fewer have done so for toddler-aged children [91]. Limited research on higher-order RRBIs is partly due to the developmental trajectory of said behaviors, which are known to decrease over time as a child matures [95,96]. Despite this possibility of inflexibility changing over time, the impact of inflexible behaviors contributes to high levels of stress in families [20,93,97]. In the current study, empowering parents to change their child’s behaviors resulted in more positive parent–child interactions. The child participants in this study were awaiting a diagnostic evaluation. Outside of this intervention, they were not receiving any specialized intervention related to the symptoms of ASD, and, for most, not receiving any intervention. Providing parents with a skill set to decrease challenging behavior (i.e., inflexibility) can improve parents’ quality of life and self-efficacy. Given the insufficiency of intervention opportunities for young children around RRBIs for those at-risk and with ASD, this study informs practice and future research in the means to address vital intervention options for families.

## Figures and Tables

**Figure 1 jcm-10-01654-f001:**
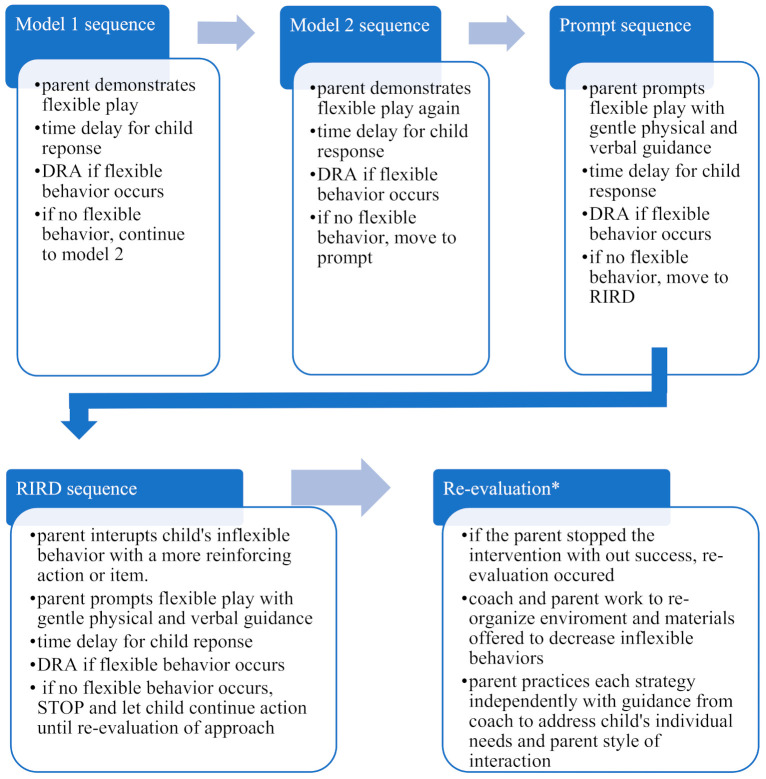
Sequence of Strategies During Intervention. The following sequences were possible: Model 1 sequence = (model 1); Model 2 sequence = (model 1 + model 2); Prompt sequence = (model 1 + model 2 + prompt); RIRD sequence = (model 1 + model 2 + prompt + RIRD). Re-evaluation* was planned for but not used. PI acted as coach in this study.

**Figure 2 jcm-10-01654-f002:**
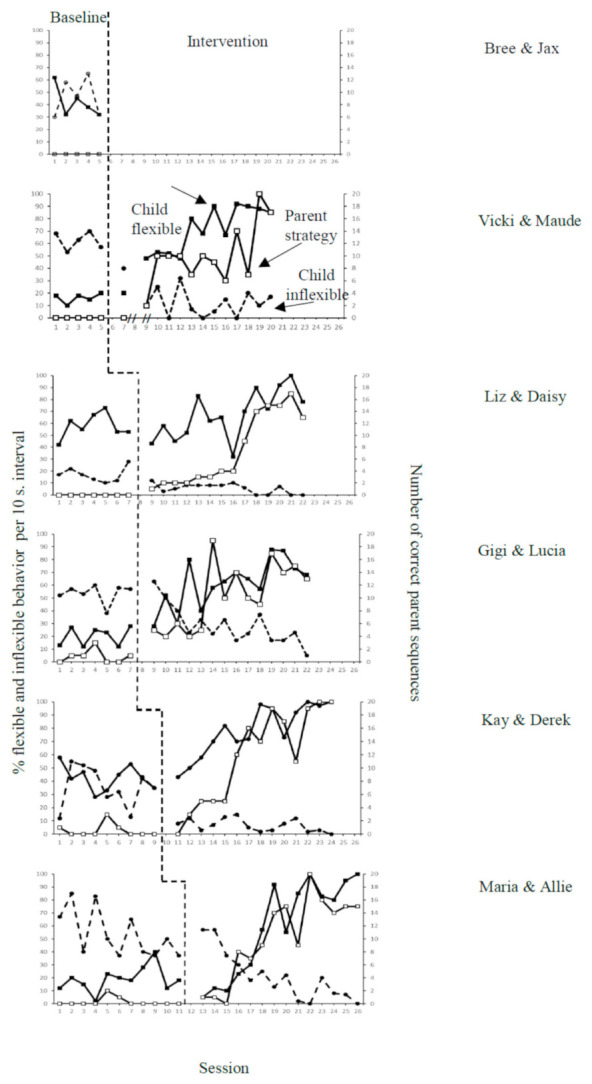
Parent strategy use, child flexible and inflexible behavior.

**Table 1 jcm-10-01654-t001:** Child Participant Characteristics.

Child Name (Dyad #)	Age in Months	Sex	Race	Alternative Diagnosis	Focus Behaviors
Jax (dyad 1)	21	M	White	Developmental Delay	1, 2, 3, 4, 5
Maude (dyad 2)	33	F	White	Developmental Delay	4, 5, 7
Daisy (dyad 3)	30	F	White	Hydrocephalus Epilepsy Hearing Impairment Developmental Delay Motor Impairment (Clubfoot)	2, 5, 8
Lucia (dyad 4)	35	F	Hispanic/Latinx	Developmental Delay	1, 6
Derek (dyad 5)	35	M	Indigenous North America/Alaskan Native	None	1, 2, 3, 4, 5
Allie (dyad 6)	31	F	Hispanic/Latinx	Developmental Delay	1, 3, 4, 5, 9

Note: Focus behaviors were from the RBS-EC pre-intervention measure completed by the parent. Although each child had target behaviors to inform their level of higher-order RRBI, the measurement for all children was more broadly categorized as inflexible and flexible behaviors. Focus behaviors: 1: upset if interrupted; 2: inflexible routine; 3: limited and intense interests; 4: fixation with parts of objects; 5: sensory seeking behaviors; 6: lines up or arranges toys and other objects; 7: narrow pre-occupation or repetitive interest with one type of toy, 8: attachment to object, 9: mouthing and carrying objects [60].

**Table 2 jcm-10-01654-t002:** Parent Participant Characteristics.

Parent Name (Dyad #)	Age in Years	Race	Marital Status	Qualifying Category
Bree (dyad 1)	25	White	Single	UR, US
Vicki (dyad 2)	30	White	Single	US
Liz (dyad 3)	25	White	Single	R, UR, US
Gigi (dyad 4)	43	Hispanic/Latinx	Married	US
Kay (dyad 5)	42	Indigenous North America/Alaskan Native	Married	UR, US
Maria (dyad 6)	42	Hispanic/Latinx	Married	US

Note: Qualifying Category: R = rural; UR = under-resourced; US = under-served.

**Table 3 jcm-10-01654-t003:** Sequence of Intervention Sessions.

Intervention Component	Component Description
Check-in & Rapport Building	Discuss how the parent and child are doing, specifics of the intervention, and areas of focus. Further descriptions of intervention strategies were discussed based on individual needs of the dyad.
Dyad Interaction(no coaching)	Parent and child interact with one another in play using areas of focus discussed during check-in. The researcher did not coach or correct here. Data collection occurred here.
Parent Coaching	The parent and researcher discuss the child’s behavior and parent responses. The researcher asked parent to practice some strategies with feedback. The researcher presented new information and reviewed past discussions as necessary. Parents were encouraged to practice strategies between meetings as necessary to reach fidelity.
Feedback	The researcher provided the parent with feedback on specific strengths and focus areas for additional practice between intervention sessions. The parent was provided with video recording of the intervention session to watch asynchronously for further learning opportunities.
Plan and Closing	The researcher and parent discussed the practice plan and goals for the upcoming week, confirming time and day for the next session.

**Table 4 jcm-10-01654-t004:** Coaching Components.

Component	Description
Joint Planning	Agreement on what actions to take during a session Understanding what caregiver will practice between visits Discussion of interactions between coach and caregiver and caregiver and child
Observation	Coach’s observation of caregiver, child, and self Caregiver’s observation of coach, child, and self
Practice	Practice skills and adjust strategy use Give feedback about practice Model, discuss, or prompt through intervention strategies
Reflection	Reflect and analyze actions and observations Reflect on skills needed for use in the next intervention
Feedback	Discuss strategies and implementation with improvements Refine skill expectations and goals

Note: All coaching components are bi-directional. The PI acted as coach in this study. The PI and caregiver both take active lead roles.

**Table 5 jcm-10-01654-t005:** Reliability of Child Behavior, Parent Strategy Use, Parent treatment Fidelity and Intervention Procedural Fidelity.

	Child Behavior M (Range)		Parent Strategy Use M (Range)				Parent Treatment Fidelity	Intervention Procedural Fidelity
	Baseline	Intervention	Overall	Baseline	Intervention	Overall	M (Range)	M (Range)
Dyad 2	87% (83–90%)	95% (80–100%)	92%	98% (95–100%)	85% (65–95%)	89%	99% (98–100%)	91% (82–100%)
Dyad 3	91% (89–92%)	88% (81–100%)	89%	98% (95–100%)	98% (95–100%)	98%	95% (88–100%)	95% (91–100%)
Dyad 4	90% (86–94%)	95% (84–100%)	94%	98% (95–100%)	88% (70–100%)	91%	93% (87–97%)	98% (91–100%)
Dyad 5	90% (85–100%)	87% (78–100%)	88%	97% (90–100%)	90% (75–100%)	93%	99% (96–100%)	98% (91–100%)
Dyad 6	88% (79–100%)	94% (82–100%)	92%	97% (90–100%)	86% (75–95%)	89%	98% (95–100%)	96% (91–100%)

Note: No data was collected for Dyad 1 due to attrition.

**Table 6 jcm-10-01654-t006:** Parent Strategy Sequence Frequency of Use and Missed Steps.

	Parent				
Descriptions	Vicki	Liz	Gigi	Kay	Maria
Strategy Sequence Use					
Model 1 sequence	81	89	113	155	46
Model 2 sequence	7	8	7	3	7
Prompt sequence	27	9	8	6	29
RIRD sequence	0	0	1	0	0
Missed Sequence Step					
DRA	10	29	12	11	9
Model in Model 2 sequence	0	2	4	0	2
Prompt in Prompt sequence	3	1	2	0	2

Note. Strategy sequences are listed in Figure 1, Phase 3: Intervention.

**Table 7 jcm-10-01654-t007:** Non-Experimental Measure Outcomes: Child Assessments.

	Participant											
Non-Experimental Measure	Bree/Jax Dyad 1		Vicki/Maude Dyad 2		Liz/Daisy Dyad 3		Gigi/Lucia Dyad 4		Kay/Derek Dyad 5		Maria/Allie Dyad 6	
	Pre	Post	Pre	Post	Pre	Post	Pre	Post	Pre	Post	Pre	Post
Assessments												
RBS-EC	2.6	---	1.9	0.6	2.2	2.2	0.4	0.4	1.9	1.6	1.5	2.2
Scale I	2.6	---	2.8	1.1	2.6	2.0	0.4	0.7	1.6	1.7	3.0	2.8
Scale II	2.7	---	0.3	0.1	2.0	1.8	0.9	0.8	2.9	2.2	1.0	2.0
Scale III	2.9	---	3.3	0.9	2.9	2.3	0.1	0.0	2.1	2.0	2.0	3.9
Scale IV	2.1	---	1.3	0.1	1.6	2.3	0.0	0.0	0.6	0.3	0.0	0.0
M-CHAT- R/F	4	---	11	6	7	12	7	6	12	8	6	5

Note: RBS-EC Scale: I. Repetitive Motor, II. Ritual & Routine, III. Restricted Interests and Behavior, IV. Self-Directed Behavior. RBS-EC score: 0 = does not occur, 1 = occurs weekly or less, 2 = occurs several times a day, 3 = occurs about daily, 4 = occurs many times a day. M-CHAT-R/F: 0–2 = low risk, 3–7 = moderate risk, 8–20 = high risk. See discussion for more details about scale results.

## Data Availability

Data available on request due to privacy restrictions. The data presented in this study are available on request from the corresponding author.

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
