# Peer review of "Coaching via Telehealth: Caregiver-Mediated Interventions for Young Children on the Waitlist for an Autism Diagnosis Using Single-Case Design"

_jcm, 2021, doi:10.3390/jcm10081654_

Round 1

Reviewer 1 Report

I thought that the text at times was wordy and difficulty to read. Maybe additional tables (e.g., on the Reliability results) would have been more concise and easier to absorb. The procedures section was also not entirely clear. Again, maybe a table/diagram with it laid out more clearly would help, especially if this may be a study that is repeated or shared with others to implement. I would have liked more information on the social validity assessments (i.e., of treatment and telehealth), especially, given the telehealth avenue for intervention delivery. There was also a disconnect between the intervention, which was implemented during and addressed playtime, and the pre- and post-assessments, which measured the child all of the time (i.e., during playtime and typical routines). Also, I am not sure that the MCHAT-R/F was the most sensitive measure to use pre- and -post intervention (limit on measures available?). These outcome measures, which are a hallmark to SCED, may have needed to be modified or the intervention needed to include a coaching element on generalization of skills (or, they needed to address that more clearly in the text). At the end of the article, the authors mention that the children were not receiving any specialized intervention related to the symptoms of ASD and for the most part not receiving any intervention (856); however, this information would have been helpful at the beginning and if it was presented more objectively as I wondered what else could be effecting the results. Also, the outcome of the ASD evaluations (and did any participants receive an evaluation during the intervention as it alluded to it?) and a post- post-assessment would have been helpful to see how the intervention lasted. Additional limits include wanting to see the authors address the impact of the intervention on other areas of ASD.

Author Response

We sincerely appreciate the feedback, questions, and comments by Reviewers 1. We have responded to each comment individually in the document uploaded as “Response to Reviewers”. This table lists comments from the reviewer as well as our response and changes in the manuscript with line numbers. The line numbers listed refer to the manuscript with track changes present to allow each reviewer and editor to see the changes made.

Thank you for allowing us the opportunity to make minor revisions to our manuscript. We wholeheartedly believe the recommendation from the reviewers have made this a better and more readable manuscript. 

Reviewer 2 Report

JCM MS. #1123468

I appreciated the opportunity to review Ms. JCM-1123468, “Coaching via Telehealth: Caregiver-Mediated Interventions for Young Children on the Waitlist for an Autism Diagnosis Using Single-Case Design.” This study evaluated the effects of training and coaching parents via telehealth to implement behavior analytic procedures with their young children who displayed repetitive behaviors, which is a characteristic symptom of autism spectrum disorder. Given that these children were waiting for autism diagnostic evaluations, and therefore receiving little to no early intervention support (presumably because of a lack of diagnosis), this study sought to close that interim gap through their training methodology. Results showed that the children’s behavior changed as a result of the procedures their parents were implementing during play-based interactions. Additionally, parental use of many of those procedures increased across time when given training and coaching via telehealth. Overall, I found this study to be very well-written, designed, and carried out. I think this study will be of interest to the reader that is looking to expand access to services when other factors traditionally limit those services (e.g., no diagnosis, no local provider, etc.). However, in looking at studies typically published in this journal, I am uncertain whether this journal is the best fit for this study. Therefore, my hesitancy in providing a recommendation is for that reason. If the editor determines that this paper’s study topic is a good fit for this journal, then I recommend that this paper be accepted contingent on revisions.

First, I suggest that the authors consider including some discussion in the introduction and discussion sections about their focus on targeting higher-order repetitive behaviors because there is currently great debate within the field of autism surrounding the value of treating these behaviors, especially in early childhood (e.g., https://www.autismspeaks.org/science-news/fda-embraces-autism-communitys-priorities-new-treatments).

Second, in the method section, I had a few questions that might be helpful to clarify:

  1. what assessments and questionnaires were completed at intake? The authors might consider listing these and providing a brief description of their purpose.
  2. were the baseline session durations 10 min total across the two visits or 10 min for each visit, resulting in 1 session per visit?
  3. were the intervention session durations 40 min total across the two visits per week or 40 min for each visit, resulting in 80 min for the week (4, 10 min sessions per visit)?
  4. during the intervention sessions, was coaching provided during the 10 min session or only provided pre- and post-session? Similarly, was coaching provided during any of the sequences or only after the end of sequence trial, whereever that sequence ended?
  5. during the prompt sequence and RIRD sequence, if hand-over-hand guidance was used, how was the time delay for the child response conducted? Was the child expected to complete the response independently after the prompt or was the prompt used only a partial prompt?

Third, in the results section, I suggest deleting the results for Bree and Jax because intervention sessions were never completed. This mother-child dyad does not add to the other results. Removing these data will also not detract from the other results. I also suggest that a few summary statements be added to the non-experimental results section.

Fourth, I had a few suggestions for the discussion section.

  1. relative to the flexibility and inflexibility of behavior section, I suggest that the authors consider how PCIT fits into their study because this model also targets the parent-child interaction through play while parents are receiving live ongoing coaching throughout the play interaction.
  2. I was really confused by the discussion regarding the need for diagnosis of autism via telehealth because this study is not focused on diagnosis. Additionally, there is research and clinical practice on diagnosis of autism via telehealth (e.g., tele-ASD-PEDS).
  3. I suggest the authors consider including a discussion about next steps for their training model in clinical practice because currently most children cannot access intervention services like these, especially from a board-certified behavior analyst, without an autism diagnosis. Thus, although this training model may be needed, it seems that many advocacy and legislative steps need to occur to open access to options such as this when children without specific diagnoses are struggling and waiting to meet criteria.

Finally, a few suggestions are provided for minor edits:

  1. in the reliability section, the intro paragraph indicates that IOA was collected for coach and parent fidelity, but the remaining paragraphs report results for IOA on child behavior and parent strategy use. Please revise accordingly.
  2. the noun-verb agreement for “data” needs to be edited throughout the manuscript from “data was” to “data were”
  3. in line 614, it appears that “flexible” needs to be edited to “inflexible”
  4. in line 657, it appears that “midpoint” needs to be added to the sentence in reference to the 5.0 rating for contextual fit

Again, I appreciated the opportunity to review this manuscript, and I hope my comments to the authors are helpful.

Author Response

We sincerely appreciate the feedback, questions, and comments by Reviewer 2. We have responded to each comment individually in the document uploaded as “Response to Reviewers.” This table lists comments from the reviewer and our response and changes in the manuscript with line numbers. The line numbers listed refer to the manuscript with track changes present to allow each reviewer and editor to see the changes.

Thank you for allowing us the opportunity to make minor revisions to our manuscript. We wholeheartedly believe the recommendation from the reviewers have made this a better and more readable manuscript. 
